Review

 

Cite this article: Russo A. 2020
Understanding the mammalian TRAP complex
function(s). *Open Biol.* **10**: 190244.

Subject Area:
biochemistry/bioinformatics/structural biology

Keywords:
endoplasmic reticulum protein translocation,
translocon-associated protein complex,
glycosylation, unfolded protein response,
endoplasmic-reticulum-associated protein
degradation, calcium-binding domain

Author for correspondence:
Antonietta Russo
e-mail: antonietta.russo@uks.eu

# Understanding the mammalian TRAP complex function(s)

Antonietta Russo

Medical Biochemistry and Molecular Biology, UKS, University of Saarland, Homburg, Germany

AR, 0000-0001-8091-4029

In eukaryotic cells, about one-third of the synthesized proteins are translocated into the endoplasmic reticulum; they are membrane or lumen resident proteins and proteins direct to the Golgi apparatus. The co-translational translocation takes place through the heterotrimeric protein-conducting channel Sec61 which is associated with the ribosome and many accessory components, such as the heterotetrameric translocon-associated protein (TRAP) complex. Recently, microscopic techniques, such as cryo-electron microscopy and cryo-electron tomography, have enabled the determination of the translocation machinery structure. However, at present, there is a lack of understanding regarding the roles of some of its components; indeed, the TRAP complex function during co-translational translocation needs to be established. In addition, TRAP may play a role during unfolded protein response, endoplasmic-reticulum-associated protein degradation and congenital disorder of glycosylation (ssr4 CDG). In this article, I describe the current understanding of the TRAP complex in the light of its possible function(s).

## 1. Introduction

In prokaryotes and eukaryotes, proteins are translocated co- and post-translationally by different pathways. In the conserved signal recognition particle (SRP) co-translation translocation, a similar protein-conducting channel translocates the nascent chains, Sec61 in eukaryotes and SecYEG in prokaryotes [1,2]. In eukaryotes, proteins and complexes that associate with the translocon (Sec61), present on endoplasmic reticulum (ER), are classified into three groups: (i) cytosolic chaperones and targeting components, such as SRP and its receptor (SR); (ii) auxiliary components, such as translocating chain-associating membrane (TRAM), translocon-associated protein (TRAP) complex, Sec62/63, ERj1, binding immunoglobulin protein (BiP), calnexin and calreticulin; and (iii) modifying enzymes, such as oligosaccharyltransferase (OST). Recently, the improved resolution of cryo-electron microscopy (cryo-EM) and cryo-electron tomography (cryo-ET) has contributed considerably to the understanding of the Sec61 structure. These three-dimensional imaging techniques allow the visualization of complexes in their physiological environment associated with native membranes when the structure does not exceed a certain thickness (0.5 µm) [3]. Sec61 spans the membranes multiple times and is made up of three different subunits—$\alpha$, $\beta$ and $\gamma$. The subunit $\alpha$ forms a channel via 10 transmembrane domains (TMDs)— five $\alpha$-helix domains in the N-terminus and five in the C-terminus connected via a short hinge helix. The subunits $\beta$ and $\gamma$ are at the periphery of the channel with one TMD and a cytosolic N-terminus (type II). According to its channel structure, Sec61 has at least two functional states: (i) the non-inserting state (15 Å) and (ii) the inserting state (diameter up 60 Å) [4]. The Sec61 achieves the open state by the nascent polypeptide moving the 'plug' inside the channel after interaction with a ribosome, and the interaction between subunits $\alpha$ and $\gamma$ (Sec61) maintains this open state. The open state can accommodate the unfolded chain or an $\alpha$-helix [5]. Inside the channel, there is also a 'pore ring', the thinnest point where six hydrophobic

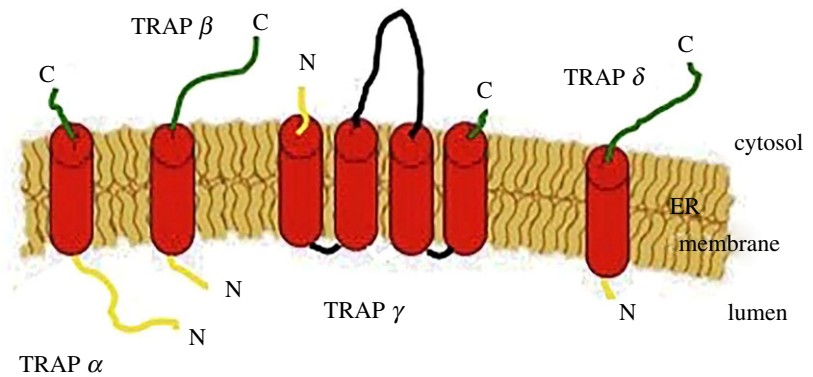

**Figure 1.** Schematic of the TRAP subunits inserted in the ER membrane: $\alpha$, $\beta$ and $\delta$ are single-spanning proteins (type I) with an SP that ranges between 17 and 23; instead, the subunit $\gamma$ has four TMDs and no SP. N-terminus in yellow, middle of the sequence in black and C-terminus in green.

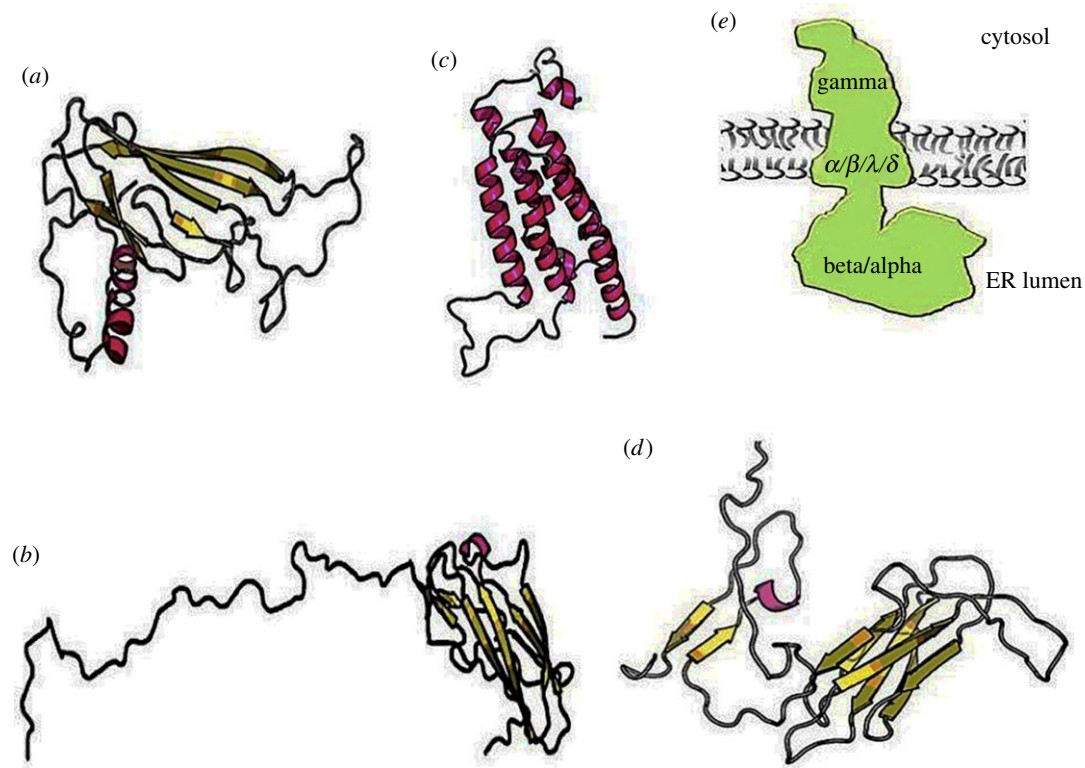

**Figure 2.** RaptorX structure prediction of TRAP subunits and structure of the complex determined by cryo-ET: (a) TRAP $\alpha$, (b) TRAP $\beta$, (c) TRAP $\gamma$ (d) TRAP $\delta$, (e) schematic of TRAP complex structure solved by cryo-ET; the subunits $\alpha$ and $\beta$ form the prominent ER luminal domain, instead, $\gamma$ has a big cytosolic domain.

residues lead to constriction during the closed state. When the plug is displaced, hydrophobic interactions are interrupted and the polypeptide with the signal peptide (SP) inserts as a loop [6–8]. The channel can open in two directions: inside (central pore) and laterally. The resident lumen proteins and proteins direct to the Golgi apparatus reach the lumen by crossing the central pore; instead, the resident membrane proteins insert in the membrane through the lateral gate (LG), a gap between two Sec61 TMDs that accommodates TMD $\alpha$-helices [9–12]. The SP has essential roles during SRP co-translational translocation: (i) ability to be recognized by the SRP; (ii) a gating step to initiate translocation, through Sec61, by the N-terminus with a pulling force; and (iii) inversion to acquire Nin-Cout orientation for cleavage [13,14]. The hydrophobicity of the SP is essential for proper protein translocation, which has been confirmed in numerous studies since the 1990s [15,16]. The translocon seems to discriminate the proteins mainly by the SP characteristics and translocating them with higher or lower efficiency [17,18]. However, the mature protein could also play a role [19,20]; indeed, proteins that are not substrates of the translocons even by adding an SP are rejected [21–23].

## 2. Structure and localization of TRAP complex

The presence of accessory structures that carry out specific function(s) during translocation is an essential aspect of translocation machinery [24]. It has been shown since the 1990s, by Blobel and co-workers [25], that in the absence of them, the protein precursors move freely into the channel and reach again the cytosolic side. Two substrate-specific auxiliary components are TRAP and TRAM [26,27]. TRAP is a ubiquitous protein complex present in all eukaryotes [28]; in mammalians, it is a heterotetrameric complex with a molecular weight of approximately 90 kDa. All four subunits, previously known as signal sequence receptors (ssr), are membrane proteins: $\alpha$ (ssr1), $\beta$ (ssr2), $\gamma$ (ssr3) and $\delta$ (ssr4). TRAP $\alpha$, $\beta$ and $\delta$ are single-spanning membrane

royalsocietypublishing.org/journal/rsob   Open Biol. 10: 190244

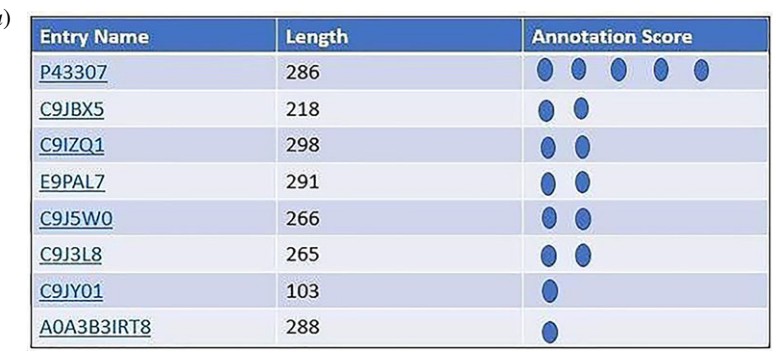

**Figure 3.** (*a*) Human TRAP α (ssr1) isoforms listed by experimental evidence (UniProt). (*b*) The mRNA alignment (Geneious) of human TRAP α isoforms (retro-translation). (*c*) The protein sequence alignment (Geneious) of human TRAP α isoforms: same N-terminus except for the shortest form which is just 103 residues long (C9JY01). The isoforms are membrane proteins type I with luminal N-term and cytosolic C-term, TMD: 208–228 (red rectangle).

(c)

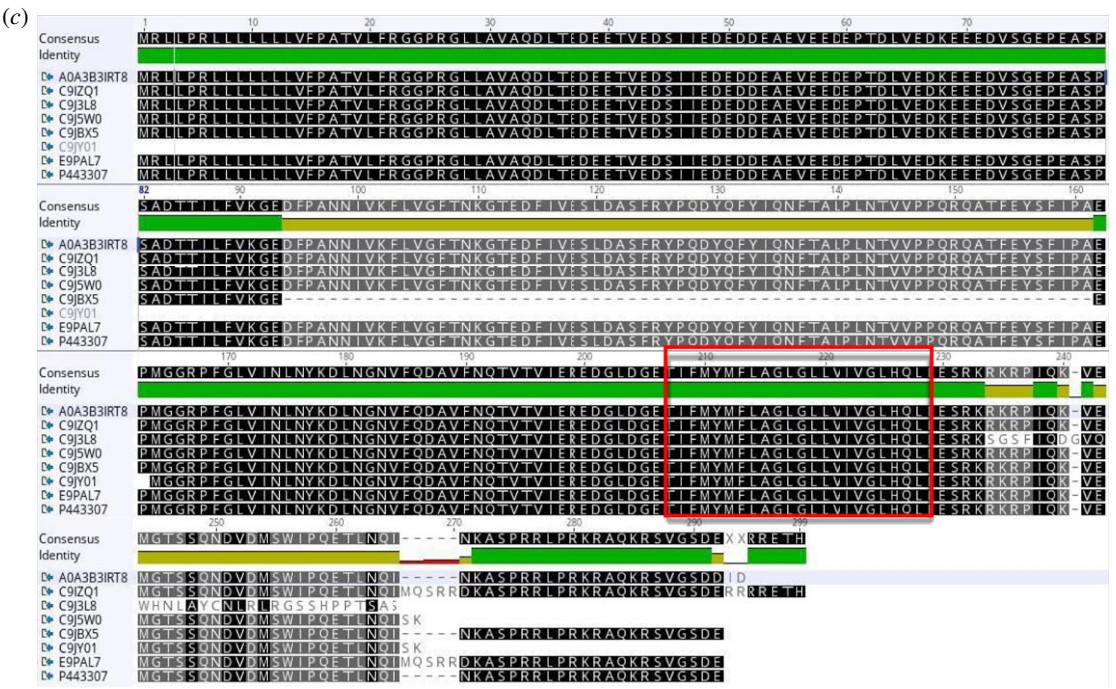

**Figure 3.** (*Continued.*)

protein type I (Nlum/Ccyt) with an SP; TRAP $\gamma$ is a multi-spanning membrane protein that crosses the membrane four times and has a conspicuous cytosolic domain and no SP [29] (figure 1). Cryo-ET methods were previously employed to compare mammalian and algae complexes (the latter and plants lack the subunits $\gamma$ and $\delta$), resulting in the determination of the TRAP complex low-resolution structure [30]; RaptorX server predicts the structure of each subunit (figure 2).

The human/*Mus musculus* TRAP$\alpha$ subunit is a glycoprotein, and the gene is located on chromosome 6 with many isoforms present, though two are more common than others. The ubiquitous general form, which is conserved between different mammalian organisms, and another form only expressed in skeletal muscle. The general form has two mRNAs, alternative polyadenylation (2.7 kb and 1.2 kb) at the 3′ non-coding regions. The mother supplies these until the eight-cell stage, then it is expressed during embryogenesis and in the adult. The other isoform is present in muscle tissue, including cardiac muscle, and is expressed after birth when the general form is turned off. The protein presents a longer C-terminus (1.8 kDa), 35% of which consists of arginine residues. Homozygous mutants die at birth for several cardiac defects. The subunit ssr1 could assist in the translocation of essential factors for heart cushion formation, such as interferon $\gamma$ ($\gamma$-INF) and atrial natriuretic peptide (ANP). These proteins inhibit the transforming growth factor-$\beta$ (TGF-$\beta$), which down-regulates the development of mesenchymal cells in endocardial cushions; this deregulation leads to mouse death [31,32]. The silencing of the TRAP $\alpha$ general isoform permits embryonic development progression because many cells are unaffected; then, the defects that arise in the heart lead to mortality.

The isoforms of human TRAP $\alpha$ (UniProt), the alignments (Geneious) of mRNA (retrotranslation) and protein sequences are shown in figure 3.

In addition to TRAP $\alpha$, the transcripts of other human TRAP subunits undergo to alternative splicing (figures 4–6)

are shown the isoforms of TRAP $\beta$ (ssr2), TRAP $\gamma$ (ssr3), TRAP $\delta$ (ssr4) and the correspondent alignments.

The human TRAP $\alpha/\beta/\gamma/\delta$ isoforms are very conserved and the alignment of the most common isoform between different mammalian organisms displays also a high identity; *Mus musculus* and the human protein sequences have 95% of identity.

The ER single-spanning membrane proteins type I may present motifs that retain the protein in the membrane, and a typical TM motif is $-K(5)X(4)K(3)X(2)X(1)$, with lysine in position $-3/-5$ in the C-terminus (K = lysine, X = any residue) [33–36]. The TRAP $\beta$ subunits of *M. musculus* and human present this retention motif at the C-terminus; it is not present in TRAP $\alpha$ and $\delta$ which are also type I membrane proteins (figure 7).

Human TRAP $\delta$ (ssr4) most common isoform presents a disulfide bridge on the ER luminal side, the cysteine residues are present in positions 3 and 34 [26] (figure 8).

The TRAP $\alpha$ subunit is crucial for mouse heart development but also other TRAP subunits are essential in some tissues during development. TRAP $\gamma$ is essential to mouse placenta formation, and the silencing of this subunit leads to embryonic organ defects in the lungs. During placenta development, many secretory proteins, such as cytokines, growth factors—FGF, PDGF, EGF and correspondent receptors, are expressed [37]. The researchers assume that ssr3 is essential to the placenta vascular network and may have a direct role in translocation, or indirectly by producing an uncoordinated TRAP complex [38]. TRAP $\gamma$ probably interacts with ribosomes via rRNA or ribosomal protein to stabilize the complex structure [39]. Moreover, TRAP $\gamma$ is necessary for *Xenopus pronephros* development [40].

Although some research has been conducted, it is not currently possible to form any conclusion regarding the role of the TRAP complex in different tissues. The knockout of one subunit leads to diverse consequences; each isoform of TRAP subunits can play a different role, or the knockout of one subunit can compromise the entire complex.

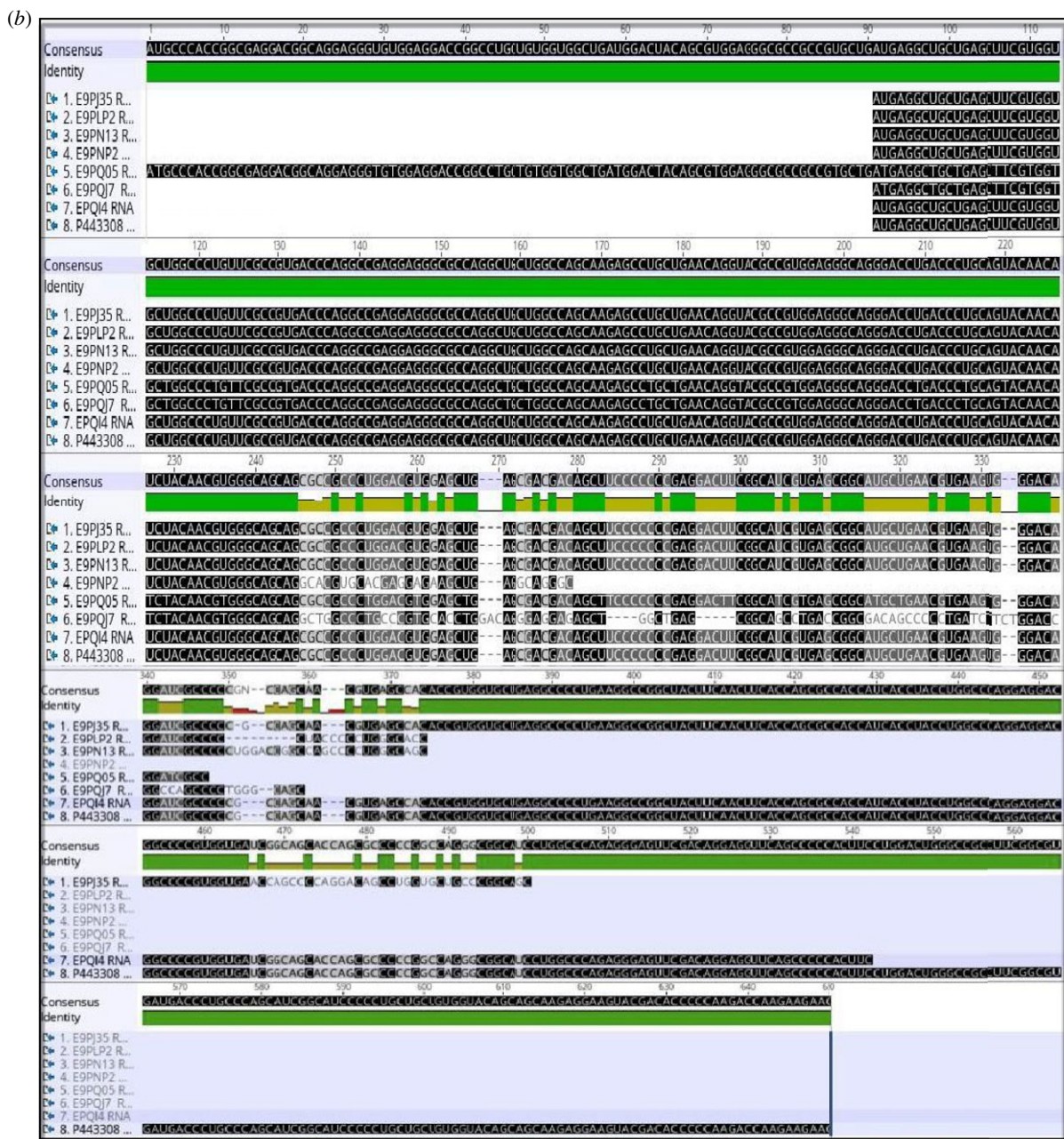

**Figure 4.** (*a*) Human TRAP β isoforms listed by experimental evidence (UniProt). (*b*) The alignment of mRNA (Geneious) of human TRAP β isoforms (retrotranslation). (*c*) The protein alignment (Geneious) of human TRAP β isoforms: except for the form with 114 amino acid residues (E9PQ05), same N terminal tail. The isoforms are membrane proteins type I with luminal N-term and cytosolic C-term, TMD: 147–167 (red rectangles).

# 3. TRAP complex and co-translational protein translocation

Systematic microscopic analyses have shown that TRAP is always present at the back of the channel Sec61, also with no-translating ribosomes [41,42]; it represents about 25% of the total volume made up of Sec61 and TRAP, and the stoichiometry between Sec61 and TRAP is 1:1. In figure 9, the comparison of the ribosome–translocon complex (RTC) with and without TRAP complex determined

royalsocietypublishing.org/journal/rsob    Open Biol. 10: 190244

*(c)*

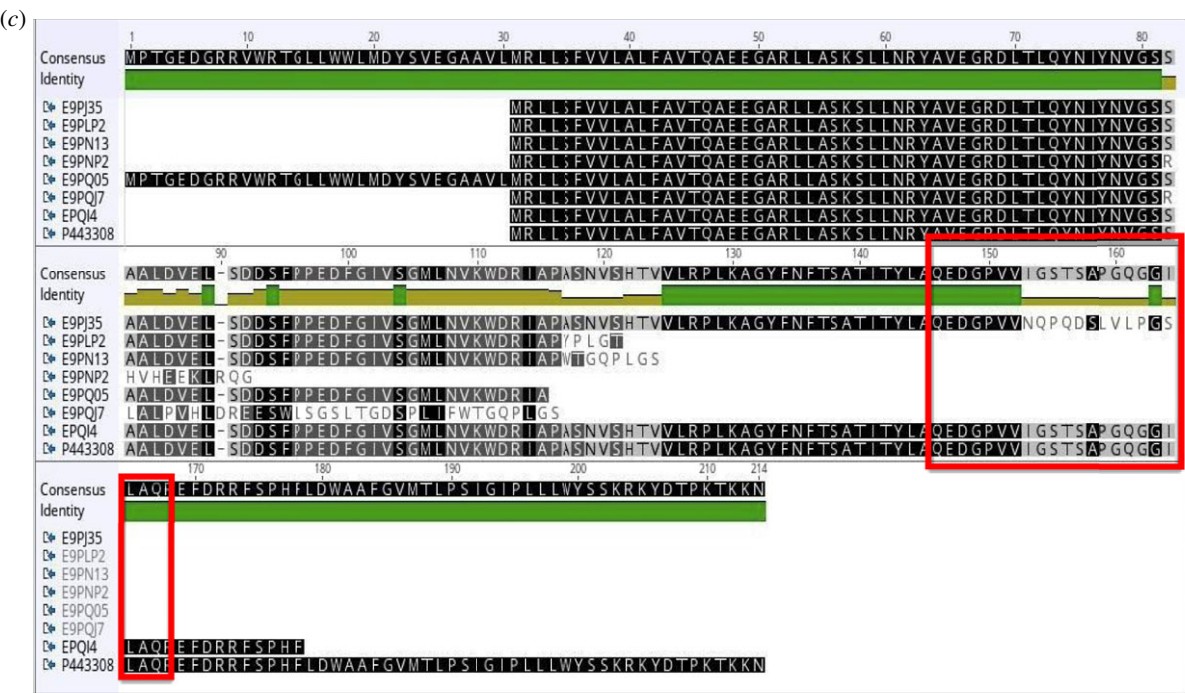

**Figure 4.** (*Continued.*)

by cryo-EM, a protuberant TRAP luminal domain is perceived.

In 2015, Pfeffer *et al.* [39] by using rough endoplasmic reticulum (rER) vesicles isolated from canine pancreases and CET/ subtomogram analysis, confirmed the structure of the ER-membrane-associated ribosomes. It was established that Sec61 is in an open state only when associated with ribosomes (RTC), TRAP is always present while OST is present in 40–70% of the complexes [39,43–45] (figure 10). It is likely that the TRAP subunits as elements of a complex interact with each other and the subunits $\alpha$ and $\beta$ interact with loop 5 of the Sec61 subunit a, proximity that has been established by the mentioned studies. TRAP $\gamma$ is very close to a ribosomal protein and this subunit has been isolated with the mammalian ribosome [46].

TRAP is a substrate-specific element of the mammalian translocon machinery. TRAP seems to be essential for substrates that have a weak SP (low hydrophobicity) and/or a high glycine and proline content [47], which is connected with the secondary structure. These residues present to the border of SP h- and c-regions, contribute to the formation of $\beta$-sheets [48]. TRAP can interact directly with the Sec61 channel to compensate a weak interaction with the substrate and maintaining an open conformation or influencing its dynamic. After interaction with ribosomes, it is plausible that different nascent proteins lead to different Sec61 conformations [6].

TRAM is also a component of the mammalian ER translocon, a membrane protein of 37 kDa, that functions as chaperone [49]; it shows substrate dependence during co-translation translocation and it has been cross-linked to substrates. It has also been demonstrated that it is involved in viral TM protein integration into the ER, each segment of the chain is before associated with Sec61$\alpha$ and then with TRAM [50]. TRAP may have similar functions and allowing the movement of the chain along the Sec61 channel after the initial force made by the ribosomes and GTP hydrolysis. The complex can also carry out a storage step until substrate

maturation; post-translational modifications, such as adding glycans [51], (hydrophilic polymers), phosphorylation (negative charges) and disulfide bridges (covalent bonds), lead to greater solubility, thermal stability and folding. This aspect is connected with the 'translocation pausing' required for the reactions of protein biogenesis. TRAP and substrate cross-linking have been detected in the late stage of translocation; this may explain the conspicuous TRAP luminal domain under the channel observed by Ménétret *et al.* [41]. TRAP complex, similarly to TRAM, could also play a role in the insertion and translocation of some membrane proteins, which is affected by the complex silencing; the authors suggest that an interaction of TRAP luminal domains with the C-terminus of nascent membrane proteins takes place [28,52]. TRAP could be specifically involved in the topology of TM proteins, they require a correct orientation when leave the LG to be accommodated in the membrane lipid bilayer. Previous investigation has demonstrated that the rapid folding in the N-terminus sequence of TM proteins and the folded structures before the signal anchor sequence restrain translocation [53]. It is plausible that the Sec61 is sufficient to translocate TM type I with cleavable SP (luminal N-terminus), but not TM proteins with a signal anchor [54].

Specifically, TRAP directly interacts with the translocon or substrate to carry out different functions: facilitate translocation and/or maintain orientation/structure of the nascent chain. Similar roles have already been demonstrated for BiP: the opening of Sec61 by nucleotide exchange [55], closure of Sec61 channel by interaction with loop 7 Sec61$\alpha$ [56] and binding to the nascent polypeptides in transit to complete translocation on ATP-dependent manner [57].

## 4. TRAP complex and glycosylation

Approximately 90% of the secretory and membrane proteins are N-glycosylated; glycosylation is the most common protein

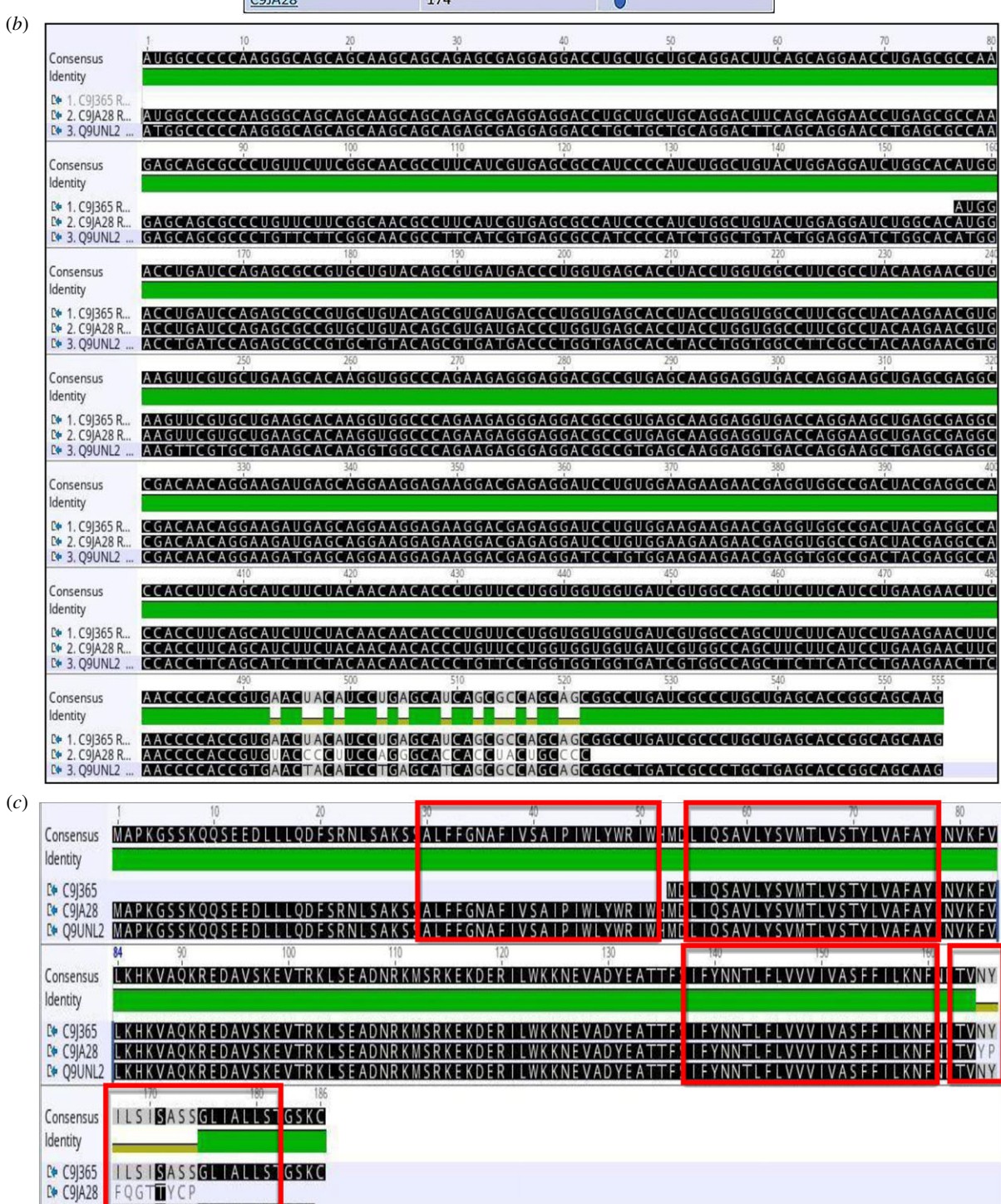

**Figure 5.** (*a*) Human TRAP γ isoforms listed by experimental evidence (UniProt). (*b*) The mRNA alignment (Geneious) of human TRAP γ isoforms (retrotranslastion). (*c*) The protein sequence alignment of human TRAP γ isoforms points out a protein (133 residues) with a shorter N-terminus (C9J365). The isoforms are multi-spanning proteins: cytosolic N-term, luminal C-term, sequence 77–138 is cytosolic; TDMs: 30–51, 55–77, 138–160, 163–182 (red rectangles).

modification in eukaryotes and directly affects protein folding positively [58]. OST, a multimeric complex of about 200 kDa, catalyses the N-glycosylation into the ER lumen [59]. The complex is part of the RTC, it is near to Sec61, ribosome 80S subunit and TRAP complex [43,60].

The TRAP δ (ssr4) subunit is associated with a congenital disorder of glycosylation (ssr4 CDG) wherein the X-linked

SSR4 gene is mutated. In the fibroblasts of these patients, the proteins are under-glycosylated and the overexpression of ssr4 partially recovers glycosylation [61]. In 'ssr4 CDG', the non-glycosylated proteins induce ER stress, but the endoplasmic-reticulum-associated degradation (ERAD) response is reduced because of the lower expression of the TRAP subunits [62]. The complex may interact with some OST

royalsocietypublishing.org/journal/rsob   Open Biol. **10**: 190244

(a)

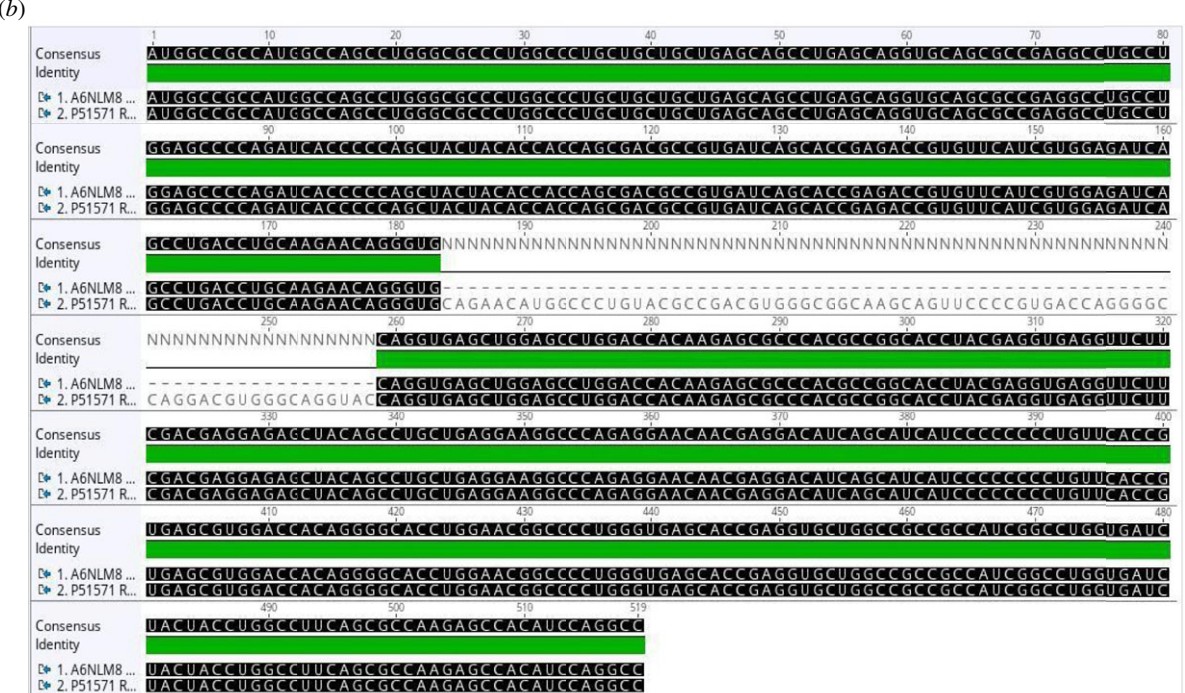

(b)

(c)

**Figure 6.** (a) The two human TRAP δ (ssr4) isoforms are listed by experimental evidence (UniProt). (b) The mRNA alignment (Geneious) of human TRAP δ (ssr4) isoforms (retrotranslation). (c) The protein sequence alignment (Geneious) of human TRAP δ (ssr4) isoforms points out lack of middle sequence in the isoform of 148 residues (A6NLM8), the remaining alignment matches 100%. The isoforms are membrane proteins type I with luminal N-term and cytosolic C-term, TMD: 145–165 (red rectangle).

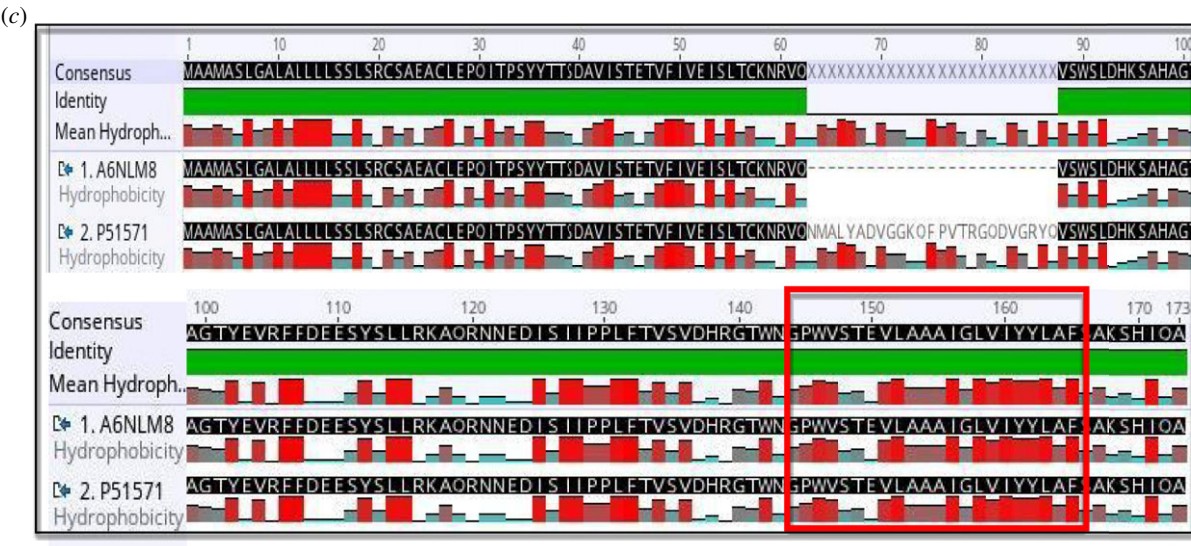

subunits, and the lack of cooperation leads to OST dysfunctionality. These interactions may modify the OST kinetic properties; indeed, STRING, a database for prediction of protein–protein interaction, predicts an interaction between TRAP δ and the DDOST subunit of OST which is essential for complex stability. Otherwise, TRAP may maintain the newly synthesized chain in a linear structure to permit N-glycosylation. Another hypothesis is that OST is a TRAP client and its synthesis is compromised. The role of TRAP δ in this congenital disorder of glycosylation and whether it plays a direct or secondary role is not currently known.

Plants and fungi lack the TRAP γ and δ subunits, yet have a coordinated complex.

## 5. TRAP complex, UPR and ERAD

The synthesis of proteins has mechanisms of quality control during different steps, such as transcription, translation, folding and assembly. In the ER, chaperones and foldases ensure the correct folding of the translocated proteins; the former prevents aggregation and the latter performs the folding steps. The

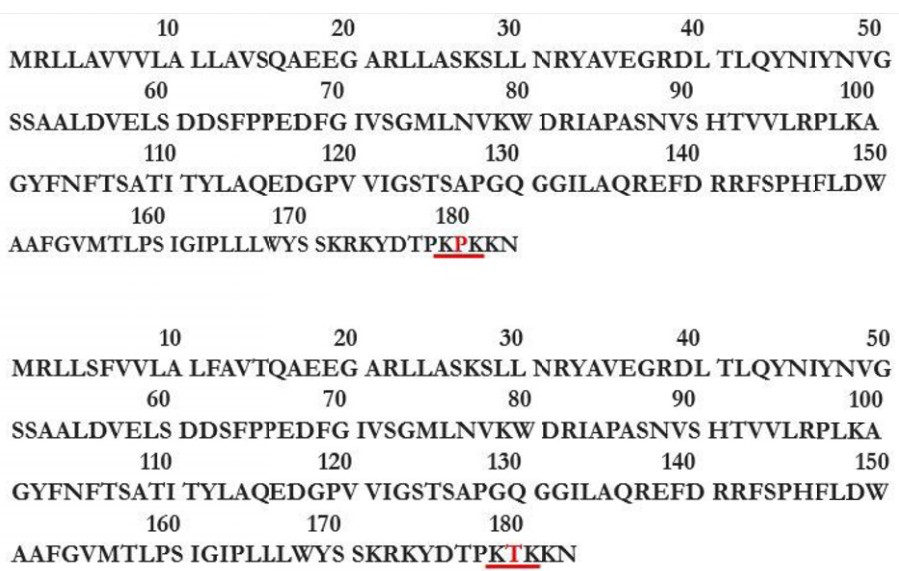

**Figure 7.** Retention TM motif—K(5)X(4)K(3)X(2)X(1)—present in the C-terminus of TRAP β *M. musculus* (above) and human (below). The difference between them is a proline in position 4 (*M. musculus*) instead of threonine (human).

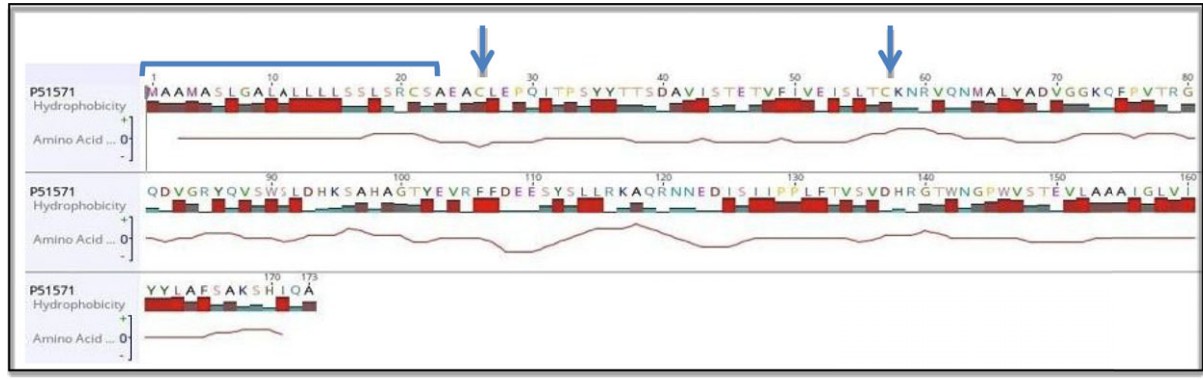

**Figure 8.** Human TRAP δ protein sequence most common isoform (P51571): two cysteine residues in the luminal domain (N-term) of the mature protein form the disulfide bridge, 3 and 34 residues (arrows). SP (bracket).

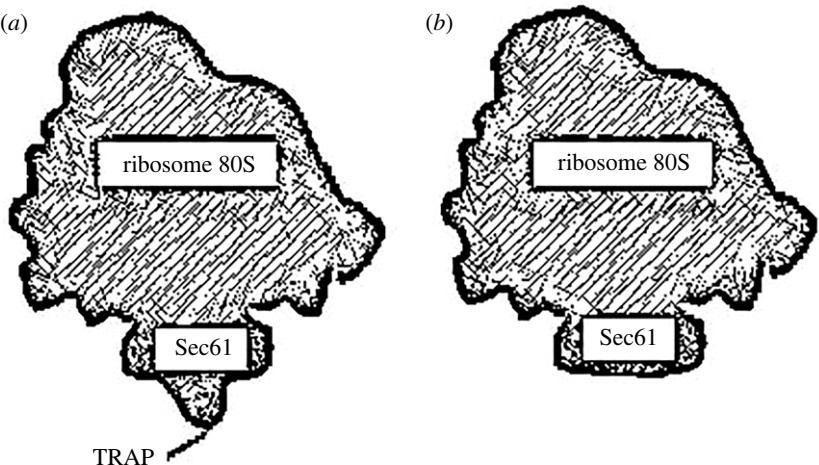

**Figure 9.** Schematic of the RTC addressed by cryo-EM: (*a*) the frontal view of the ribosome, Sec61 and TRAP under the channel; (*b*) frontal view of the ribosome and Sec61.

recognition of unfolded proteins activates the unfolded protein response (UPR), which leads to the ERAD pathways [63]. The proteasome, a prominent structure, degrades the proteins after the attachment of multiple copies of ubiquitin (protein hydrolysis), [64,65]. Three main steps are necessary for ERAD: (i) recognition and targeting, (ii) retro-translocation and ubiquitination, and (iii) proteasome targeting and degradation.

The UPR also occurs due to the production of proteins overcoming the necessity of the cell; the nascent proteins

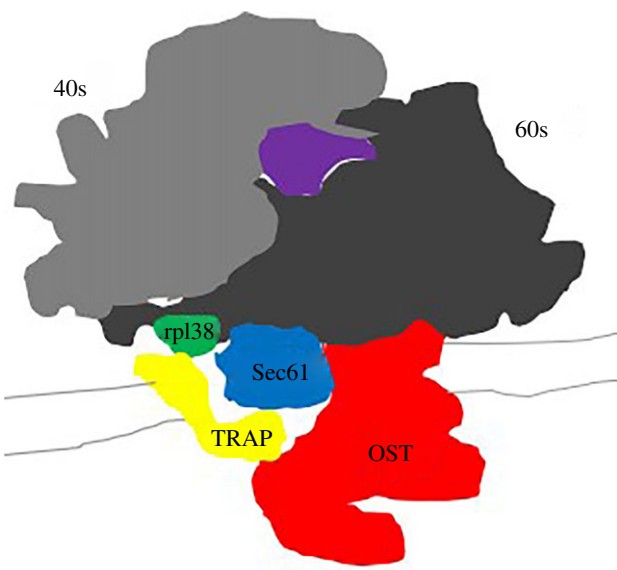

**Figure 10.** Schematic of the ER-membrane-associated ribosome determined by cryo-ET: Sec61 (blue), TRAP (yellow) and OST (red). It is visible a prominent ER luminal domain of TRAP ($\alpha$ and $\beta$) under the channel Sec61 and a cytosolic domain of TRAP ($\gamma$) close to the subunits 60s of the ribosome and precisely to the ribosomal protein rpl38 (green). TRAP complex is also close to the OST complex which interacts with the translocon and the ribosome.

misfold and aggregate because of their high concentrations ($300-400 \text{ g l}^{-1}$), [66].

Moreover, the perturbation of any other process in the ER, such as transport/phospholipid synthesis, and distribution/calcium storage, drives ER stress and UPR [67,68]. The ubiquitous ER membrane proteins in UPR are Ire1$\alpha$, PERK and ATF6; three different pathways that stimulate transcription factors to express the ER chaperones and ERAD components. The upregulation of folding, degradation and the downregulation of protein synthesis alleviate stress.

Whether the TRAP complex can carry out more than one function remains to be elucidated. Some findings suggest a role of the TRAP complex in the UPR pathway and cellular equilibrium [62]. TRAP interacts with unfolded substrates but not with the wild-type form such as superoxide dismutase (SOD1) [69]; there is TRAP expression induction under ER stress by IRE $\alpha$ pathway, also under GM-GST stimulation which leads to the transcription of many genes, UPR and ERAD [70]. Some proteins, such as calreticulin, calnexin and BiP, have a role in folding and quality control; this could be the case for TRAP. Some studies have suggested that Hrd1 and Hrd3 retro-translocate abnormal proteins after ubiquitination [71]. Nonetheless, the translocon Sec61 could retro-translocate the proteins that undergo degradation [72,73]. Indeed, it interacts with ERAD substrates and the proteasome [74]. The retro-translocon Sec61 may require support from the luminal side and the overexpression of TRAP subunits during ERAD could be connected with these processes. However, which associated components push the substrate through the channel is currently unknown. It is also not known whether ubiquitination is sufficient.

## 6. TRAP complex and calnexin

Calnexin (90 kDa) is a membrane protein type I, like TRAP$\alpha$, and probably both bind calcium in the ER lumen [75]. The TRAP subunit ssr1 has a non-canonical EF domain in the N-terminus. Remarkably, calnexin is also a component of the RTC and, like TRAP, is close to the translocating polypeptide; calnexin captures some substrates that acquire N-linked glycans. The palmitoylation of calnexin by DHHC6 permits the interaction with TRAP $\alpha$. The palmitoylation also recruits the actin cytoskeleton needed for RTC stabilization [76] (figure 11). Similar to the TRAP subunits, calnexin is involved in the ERAD pathways and the cnx$-/-$ cells have active UPR for acute stress [77]. In addition, calnexin plays a role in protein folding [78,79].

## 7. TRAP $\alpha$ non-canonical EF-hand motif

Calcium has regulatory and structural roles and is a crucial element both outside and inside cells. Outside the cells, its concentration is approximately $10^{-3}$ M, while inside the cells, it is 104 folds lower and mainly concentrated into the ER. Many proteins bind $Ca^{2+}$ to maintain/change their structure and carry out biological functions. The common motif DXDXDG is included in a linear sequence of about 30 amino acid residues, where two perpendicular $\alpha$-helices form the 12-residue $Ca^{2+}$-binding loop; the binding residues are in positions 1, 3, 5, 7, 9 and 12, with the latter, always being Glu (E) or Asp (D), which are negatively charged residues that interact with the positively charged $Ca^{2+}$ [80]. These canonical EF-hand domains are located in calmodulin proteins; there are also non-canonical EF-hand domains or EF-hand-like domains that are mostly present in the N-termini of S100 and S100-like proteins.

By aligning TRAP$\alpha$ ubiquitous isoform (most common isoform) protein sequence with the human $\alpha$-Palvalbumin (Parv) [81] non-canonical EF-hand motif, it is almost clear that this motif is present in the N-terminus of *M. musculus* and Human proteins (figure 12). Therefore, TRAP $\alpha$ may have a calcium-binding role in the interaction with the complex Sec61; TRAP can undergo different conformation that influences its interactions. Previous studies report that the binding of calcium by the C-terminal EF-hand domain of Se62 leads to the dissociation from N-terminus of its interacting partner Sec61 [82].

## 8. Concluding remarks

The ER co-translational protein translocation relies on general structures: targeting signals, membrane receptors, transmembrane channels and accessory components. It is not currently known when some accessory components are necessary and the channel is insufficient; the functions of these components require further study. The field limitation is the analysis of subcellular structures during their function. Additionally, separate components from cell fractions require appropriate representation and conditions. Undoubtedly, methods such as cryo-EM/ET are appropriate for structural analysis in entire cells or lysates, and they have been extensively used to study the TRAP complex. However, the assembling of the subsequent snapshots to describe the entire biological mechanism is a major disadvantage. The processes are rapid and consist of real dynamics; for instance, the configuration between the RTC and nascent polypeptide changes overtime; it is necessary to overcome these weaknesses to

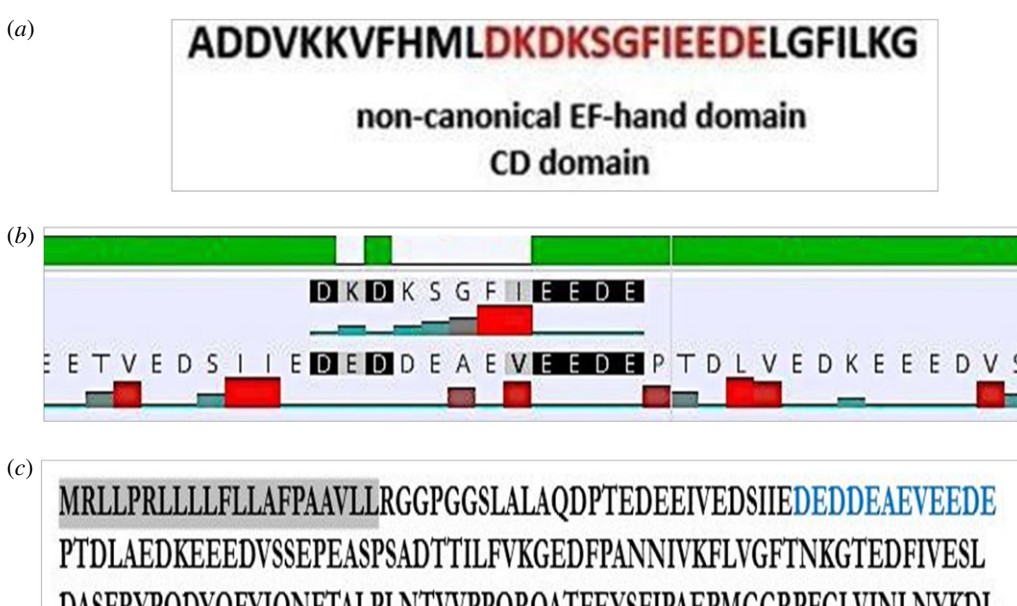

**Figure 11.** Calnexin-like ssr1 is a component of the RTC, both are close to the translocating polypeptide. The interaction of calnexin with TRAP$\alpha$ depends on Calx palmitoylation by DHHC6 which recruits also the actin cytoskeleton.

(*a*)

**ADDVKKVFHMLDKDKSGFIEEDELGFILKG**

non-canonical EF-hand domain
CD domain

(*b*)

DKDKSGFIEEDE

EETVEDSIIEDEDDEAEVEEDEPTDLVEDKEEEDVS

(*c*)

MRLLPRLLLLFLLAFPAAVLLRGGPGGSLALAQDPTEDEEIVEDSIIEDEDDEAEVEEDE
PTDLAEDKEEEDVSSEPEASPSADTTILFVKGEDFPANNIVKFLVGFTNKGTEDFIVESL
DASFRYPQDYQFYIQNFTALPLNTVVPPQRQATFEYSFIPAEPMGGRPFGLVINLNYKDL
NGNVFQDAVFNQTVTVIEREDGLDGETIFMYMFLAGLGLLVVVGLHQLLESRKRKRPIQ
KVEMGTSSQNDVDMSWIPQETLNQINKASPRRQPRKRAQKRSVGSDE

(*d*)

MRLLPRLLLLLLLVFPATVLFRGGPRGLLAVAQDLTEDEETVEDSIIEDEDDEAEVEEDE
PTDLVEDKEEEDVSGEPEASPSADTTILFVKGEDFPANNIVKFLVGFTNKGTEDFIVESL
DASFRYPQDYQFYIQNFTALPLNTVVPPQRQATFEYSFIPAEPMGGRPFGLVINLNYKDL
NGNVFQDAVFNQTVTVIEREDGLDGETIFMYMFLAGLGLLVIVGLHQLLESRKRKRPIQK
VEMGTSSQNDVDMSWIPQETLNQINKASPRRLPRKRAQKRSVGSDE

**Figure 12.** (*a*) Non-canonical EF-hand domain present in human $\alpha$-Palvalbumin (Parv). Parvalbumin is a calcium-binding protein involved in intracellular calcium signalling. CD, calcium-binding. (*b*) The alignment between TRAP$\alpha$ sequence and the non-canonical EF-hand domain (Parv); red bars: hydrophobicity. (*c*) The entire *M. musculus* TRAP$\alpha$ sequence (most common isoform) and the probable non-canonical EF-hand domain (blue) in the N-terminus. (*d*) The sequence of human TRAP $\alpha$ (most common isoform) and the probable non-canonical EF-hand domains (blue) in the N-terminus. SP highlighted in grey.

royalsocietypublishing.org/journal/rsob Open Biol. 10: 190244

**Table 1.** Summary of processes, TRAP expression, possible TRAP function(s) and effects.

| processes | TRAP expressions | TRAP functions | effects |
|---|---|---|---|
| co-translation protein translocation [26,28–30,32,38,40–42,47] | Sec61 and TRAP stoichiometric ratio 1 : 1 | – assists Sec61 open state<br>– chaperone: interaction with substrate | substrate-dependent |
| unfolded protein response (UPR) [62,69,70] | TRAP overexpressed | – chaperone: protein maturation<br>– recognition of unfolded proteins | ↑UPR |
| ER-associated protein degradation (ERAD) [62,69,70] | TRAP silencing | – secondary effect of UPR<br>– retro-translocation | ↓ERAD |
| protein glycosylation [61] | – TRAP $\delta$ absent<br>– TRAP $\alpha$, $\beta$, $\gamma$ underexpressed | – chaperone: assists protein modification<br>– interaction with OST<br>– OST substrate of TRAP | ssr4 CDG: under-glycosylated |

be able to describe the entire translocation processes across the ER membrane and answer to many open questions.

By following the literature, it seems that some roles of TRAP complex are redundant with BiP, TRAM and calnexin. What is the exact TRAP contribution? Overall—Does TRAP interact with the substrate or the translocon to carry out its function?

What is the function of TRAP $\alpha$ calcium-binding? The presence of non-canonical EF-hand domain suggests that this subunit binds $Ca^{2+}$. This binding could change the conformation of TRAP $\alpha$ to control a transient interaction or merely increase rigidity in a stable physical interaction.

Further analysis will also be necessary to determine if some interactions between the TRAP and OST complexes take place. Reduced glycosylation in a congenital disorder is linked to the absence of TRAP $\delta$ and reduced expression

of other TRAP subunits. Together, these results lead to some conclusions: lack of TRAP complex stability/function and/or lack of interaction between TRAP and OST.

Why the silencing of TRAP subunits leads to reduced ERAD?

In summary, although some studies have been carried out regarding the TRAP complex (table 1), no single study exists that adequately addresses its role inside the ER and during protein translocation. Further integration of many uncoordinated and divergent studies is necessary. This integration could establish the molecular functions and biological processes beyond the knowledge of the cellular component and structure.

Data accessibility. This article has no additional data.

Competing interests. I declare I have no competing interests.

Funding. I received no funding for this study.

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
