## [Reviewer comments · Open Biology]

Review History

RSOB-19-0244.R0 (Original submission)

Review form: Reviewer 1

Recommendation

Reject – article is not of sufficient interest (we will consider a transfer to another journal)

Do you have any ethical concerns with this paper?

No

Comments to the Author

In this manuscript the author reviewed the current knowledge of TRAP complex, an heterotetrameric complex associated to the translocon. The author revised the structural information that appeared in the last years, basically thanks to cryo-EM and cryo-ET data, and link this information to possible functions of the TRAP complex. General comment followed by specific comments/criticisms are below.

General:

The review includes important errors and omissions that significantly complicates reading. Figures are poorly informative and with very low quality (at least as included in the revised pdf).

The author stated as Tab. (Tables, I assume) images than to me look more like 'Figures' than actual 'Tables'.

Tab. 6 is missing.

In the body text it is mentioned up to Tab. 12, while the revised file only contains up to Tab. 8.

References need profuse curation.

Abbreviations list is required.

Specific:

1- Abstract. The first sentence can be misinterpreted. Together, translocated into the ER lumen and ER inserted proteins represent about one-third of the proteome. As written, it is a bit confusing, please clarify.

2- Abstract. The following sentence is either difficult to understand or incorrect: "...however, research has consistently shown a lack of understanding about the role(s) of some of its components. TRAP complex is involved in co-translational translocation, unfolded protein response, ERAD pathways, and glycosylation disorder (ssr4 CDG), but its role is not unknown." Please rephrase.

3- Introduction (page 1). The following sentence need a reference to justify the data provided: "According to its channel structure, the Sec61 has at least two functional states: no- inserting state (9-15 Å) and inserting state (diameter 40-60 Å)."

4- Introduction (page 2). Either explain the differences between TRAM1 and TRAM2 or remove the numbers in "TRAM1/2" and refer to this protein as TRAM.

5- Introduction (page 2). I cannot understand what the author wants to say in the following sentence: "...It is known that there is an interconnection between the SP and the mature protein (21, 22) and the SP is functionally distinct and optimized for the mature protein sequence, in fact, no-clients of the translocon even by adding a SP are rejected (23-26)."

6- Fig. 2: explain what the yellow, green and black lines represent in the legend.

7- Fig. 3 is not informative. Please, produce a caption of the TRAP complex using the appropriate PDB codes.

8- Tab. 1-6 appear to be just screenshots (webpages should be mentioned), very low quality and poorly informative images. At least the transmembrane regions should be highlighted, as well as the cytosol and ER lumen exposed loops.

9- Page 5, last paragraph is dedicated to type I membrane proteins, please mention 'type I' explicitly. At the end of the paragraph it should be referred to Tab. 5.

10- Fig. 4 is a cartoon, if the author wants to show the structure, an appropriated caption should be provided, for instance to highlight rpl38 protein as mentioned both in the text and in the figure legend.

11- Page 8: TRAM has also been crosslinked to nascent chains with more than 100 residues. Then, this fact cannot be argued as a differential feature between TRAM and TRAP. Actually, TRAM deserves a clearer description.

12- Page 8: "TRAM also by crosslinking experiments, seems to be involved in viral TM protein integration into the ER, each segment of the chain is before associated with Sec61 α and then with TRAM (44)."

Ref. 44 that deals with TRAP is not correct, the experiments described where published elsewhere. Please, place the appropriate reference.

13- From reference 73 up to the end the literature is totally inconsistent, and its correctness cannot be checked. Please, revise it carefully.

14- Figs. 6 and 7, again with poor quality, they are not very informative. If maintained, both should be significantly improved, and STRING consortium acknowledged (Fig. 6).

15- Page 11. The non-canonical EF-domain at the N-terminus of TRAPalpha is not shown in Tab. 6, which actually is devoted to TRAPdelta, but Tab. 7.

16- Last paragraph: where are Tab. 11-12?

17- Page 12. Sec62 should be introduced and properly spelled (line 2).

18- Page 13 last paragraph it refers to Tab. 8, I guess.

Review form: Reviewer 2

Recommendation

Major revision is needed (please make suggestions in comments)

Do you have any ethical concerns with this paper?

No

Comments to the Author

The manuscript covers a topic that would be interesting to a general readership. However in its present form it is of insufficient quality.

Comments:

According to title and abstract, the aim of the manuscript is to discuss the function of the mammalian TRAP and therefore it refers generally to the mammalian translocon with its central component, the Sec61 complex. However, many citations refer to work done with prokaryotic organisms. This may be confusing for readers not familiar with the field, especially as the functional homology between the translocon in the mammalian ER and the translocon in the plasma membrane in prokaryotic organisms was not introduced.

In several cases, statements are supported not by citing original literature but other reviews. In other cases the citation is not directly linked to the information provided.

Also several citations refer to parts of the discussion section in a publication and not to the original data without making this point clear for the reader.

Generally, the manuscript suffers from unclear or imprecise formulations. Here are just some examples I found in first two pages:

"TRAP complex is involved in co-translational translocation, unfolded protein response, ERAD pathways, and glycosylation disorder (ssr4 CDG), but its role is not unknown."

"The proteins in the cell are synthesized at the velocity of 2 to 13 million per minute and about one-third are imported or reside in the ER; they are classified as insoluble intracellular, soluble secreted, type I and II membrane proteins and multi-spanning membrane proteins."

In other cases formulations lead to a perhaps unwanted overemphasising.

"...and the SP is functionally distinct and optimized for the mature protein sequence, in fact, no clients of the translocon even by adding a SP are rejected (23-26)."

Before publication, the author must therefore carefully re-examine the complete manuscript!

The quality of many figures is of low standard, some like Fig. 1 could be omitted, as they give no relevant information.

Page 3 - the molecular weight of the TRAP complex in mammalian cell is less than 150kDa

Citation 32 seems to contain typing errors; between cit. 81 and 84 there is confusion in the order of numbers

Table 1-4 - If one wants to discuss isoforms, one should show alignments of both, RNA and primary protein structure. This should be done with sufficient resolution of the figure!

Decision letter (RSOB-19-0244.R0)

06-Jan-2020

Dear Dr Russo,

We are writing to inform you that the Editor has reached a decision on your manuscript RSOB-19-0244 entitled "Understanding the mammalian TRAP complex function(s)", submitted to Open Biology.

As you will see from the reviewers' comments below, there are a number of criticisms that prevent us from accepting your manuscript at this stage. The reviewers suggest, however, that a revised version could be acceptable, if you are able to address their concerns. If you think that you can deal satisfactorily with the reviewer's suggestions, we would be pleased to consider a revised manuscript.

The revision will be re-reviewed, where possible, by the original referees. As such, please submit the revised version of your manuscript within four weeks. If you do not think you will be able to meet this date please let us know immediately.

When submitting your revised manuscript, please respond to the comments made by the referee(s) and upload a file "Response to Referees" in "Section 6 - File Upload". You can use this to document any changes you make to the original manuscript. In order to expedite the processing of the revised manuscript, please be as specific as possible in your response to the referee(s).

Please see our detailed instructions for revision requirements
<https://royalsociety.org/journals/authors/author-guidelines/>

Sincerely,
The Open Biology Team
<mailto:openbiology@royalsociety.org>

Reviewer(s)' Comments to Author(s):

Referee: 1

Comments to the Author(s)

In this manuscript the author reviewed the current knowledge of TRAP complex, an heterotetrameric complex associated to the translocon. The author revised the structural information that appeared in the last years, basically thanks to cryo-EM and cryo-ET data, and link this information to possible functions of the TRAP complex. General comment followed by specific comments/criticisms are below.

General:

The review includes important errors and omissions that significantly complicates reading. Figures are poorly informative and with very low quality (at least as included in the revised pdf). The author stated as Tab. (Tables, I assume) images than to me look more like 'Figures' than actual 'Tables'.

Tab. 6 is missing.

In the body text it is mentioned up to Tab. 12, while the revised file only contains up to Tab. 8.

References need profuse curation.

Abbreviations list is required.

Specific:

- 1- Abstract. The first sentence can be misinterpreted. Together, translocated into the ER lumen and ER inserted proteins represent about one-third of the proteome. As written, it is a bit confusing, please clarify.
- 2- Abstract. The following sentence is either difficult to understand or incorrect: "...however, research has consistently shown a lack of understanding about the role(s) of some of its components. TRAP complex is involved in co-translational translocation, unfolded protein response, ERAD pathways, and glycosylation disorder (ssr4 CDG), but its role is not unknown." Please rephrase.
- 3- Introduction (page 1). The following sentence need a reference to justify the data provided: "According to its channel structure, the Sec61 has at least two functional states: no- inserting state (9-15 Å) and inserting state (diameter 40-60 Å)."
- 4- Introduction (page 2). Either explain the differences between TRAM1 and TRAM2 or remove the numbers in "TRAM1/2" and refer to this protein as TRAM.
- 5- Introduction (page 2). I cannot understand what the author wants to say in the following sentence: "...It is known that there is an interconnection between the SP and the mature protein (21, 22) and the SP is functionally distinct and optimized for the mature protein sequence, in fact, no-clients of the translocon even by adding a SP are rejected (23-26)."
- 6- Fig. 2: explain what the yellow, green and black lines represent in the legend.
- 7- Fig. 3 is not informative. Please, produce a caption of the TRAP complex using the appropriate PDB codes.
- 8- Tab. 1-6 appear to be just screenshots (webpages should be mentioned), very low quality and poorly informative images. At least the transmembrane regions should be highlighted, as well as the cytosol and ER lumen exposed loops.
- 9- Page 5, last paragraph is dedicated to type I membrane proteins, please mention 'type I' explicitly. At the end of the paragraph it should be referred to Tab. 5.
- 10- Fig. 4 is a cartoon, if the author wants to show the structure, an appropriated caption should be provided, for instance to highlight rpl38 protein as mentioned both in the text and in the figure legend.
- 11- Page 8: TRAM has also been crosslinked to nascent chains with more than 100 residues. Then, this fact cannot be argued as a differential feature between TRAM and TRAP. Actually, TRAM deserves a clearer description.
- 12- Page 8: "TRAM also by crosslinking experiments, seems to be involved in viral TM protein integration into the ER, each segment of the chain is before associated with Sec61 α and then with TRAM (44)."
- Ref. 44 that deals with TRAP is not correct, the experiments described where published elsewhere. Please, place the appropriate reference.
- 13- From reference 73 up to the end the literature is totally inconsistent, and its correctness cannot be checked. Please, revise it carefully.
- 14- Figs. 6 and 7, again with poor quality, they are not very informative. If maintained, both should be significantly improved, and STRING consortium acknowledged (Fig. 6).
- 15- Page 11. The non-canonical EF-domain at the N-terminus of TRAPalpha is not shown in Tab. 6, which actually is devoted to TRAPdelta, but Tab. 7.
- 16- Last paragraph: where are Tab. 11-12?
- 17- Page 12. Sec62 should be introduced and properly spelled (line 2).
- 18- Page 13 last paragraph it refers to Tab. 8, I guess.

Referee: 2

Comments to the Author(s)

The manuscript covers a topic that would be interesting to a general readership. However in its present form it is of insufficient quality.

Comments:

According to title and abstract, the aim of the manuscript is to discuss the function of the mammalian TRAP and therefore it refers generally to the mammalian translocon with its central component, the Sec61 complex. However, many citations refer to work done with prokaryotic organisms. This may be confusing for readers not familiar with the field, especially as the functional homology between the translocon in the mammalian ER and the translocon in the plasma membrane in prokaryotic organisms was not introduced.

In several cases, statements are supported not by citing original literature but other reviews. In other cases the citation is not directly linked to the information provided.

Also several citations refer to parts of the discussion section in a publication and not to the original data without making this point clear for the reader.

Generally, the manuscript suffers from unclear or imprecise formulations. Here are just some examples I found in first two pages:

“TRAP complex is involved in co-translational translocation, unfolded protein response, ERAD pathways, and glycosylation disorder (ssr4 CDG), but its role is not unknown.”

“The proteins in the cell are synthesized at the velocity of 2 to 13 million per minute and about one-third are imported or reside in the ER; they are classified as insoluble intracellular, soluble secreted, type I and II membrane proteins and multi-spanning membrane proteins.”

In other cases formulations lead to a perhaps unwanted overemphasising.

“...and the SP is functionally distinct and optimized for the mature protein sequence, in fact, no-clients of the translocon even by adding a SP are rejected (23-26).”

Before publication, the author must therefore carefully re-examine the complete manuscript!

The quality of many figures is of low standard, some like Fig. 1 could be omitted, as they give no relevant information.

Page 3 - the molecular weight of the TRAP complex in mammalian cell is less than 150kDa

Citation 32 seems to contain typing errors; between cit. 81 and 84 there is confusion in the order of numbers

Table 1-4 - If one wants to discuss isoforms, one should show alignments of both, RNA and primary protein structure. This should be done with sufficient resolution of the figure!

Author's Response to Decision Letter for (RSOB-19-0244.R0)

See Appendix A.

RSOB-19-0244.R1 (Revision)

Review form: Reviewer 1

Recommendation

Major revision is needed (please make suggestions in comments)

Do you have any ethical concerns with this paper?

No

Comments to the Author

The author has attended some of the criticisms raised in the previous revision but still some corrections are needed. Following I will describe some specific examples but the manuscript contains more than the mentioned.

Specific:

1- Abstract. The first sentence is incorrect after the revision, since 1/3 of the proteins are not translocated or “reside” into the ER. Actually, this percentage of proteins are translocated or ‘inserted’ through the ER, but a large population can move to other membranes like Golgi, plasma,..., meaning that not all the membrane proteins inserted into the ER in fact “reside” in the ER.

2- Introduction. The first paragraph of this section is specially confusing. For instance, the first sentence is incorrect, not all prokaryotic membrane proteins are transported across the plasma membrane through Sec A. The second sentence regarding eukaryotic cells must be corrected according to what I mentioned above; and the idea behind the classification in “soluble intracellular, soluble secreted, type I/II...” makes no sense to me if it is referring to one-third of proteins that are targeted to the membrane.

The sentence (line 9) “In mammalians,...” it’s incorrect in its current form, several truncated forms of strict cotranslationally inserted proteins as short as 86-96 residues have been crosslinked to SRP and/or Sec61alpha. Examples: DOI:

10.1038/nsmb994 doi: 10.1128/JVI.00125-10.

The expression “no-clients of translocons...” in the last sentence of the introduction has no sense to me.

In general, the introduction of this review must be carefully curated, for that the author should read and cite some authorized recent revisions, like doi: 10.1146/annurev-biochem-013118-111717 and doi: 10.1515/hsz-2014-0205, to avoid incorrections or imprecise formulations related to the mechanisms of targeting and cotranslational insertion of membrane proteins.

3- Fig. 1. There is an extra loop between TM2 and TM3 in TRAP gamma that should be removed.

4- Page 14, line 7: replace beta-barrel by beta-sheet.

5- Page 14, the first sentence of the second paragraph is clearly outdated (the reference given is close to 25 years old!). A chaperoning function for TRAM has been suggested and its topology defined, doi: 10.1016/j.jmb.2011.01.009.

6- Reference 61 (page15, line 6) it’s not related to what it is mentioned in the text, please remove it.

7- Introduce STRING algorithm on page 15, not every reader should know what this algorithm does.

8- Page 16: add ‘expression’ between “TRAP induction” in the second and third points suggested for TRAP role in the UPR pathway.

9- Regarding the non-canonical EF domain in *ssr1* on page 16 add ‘(see below)’ at the end of the sentence.

10- Page 17, last sentence, please correct Sec62.

11- Tab. 7: explain panel B, what are the red, gray and green colors referred to? Why there are three sequence lines if only compares TRAP-alpha with Parv?

12- Page 19: in the sentence “The presence of non-canonical EF- hand domain almost confirms that this subunit binds Ca²⁺”, ‘suggests’ is a better choice than “almost confirms”.

Review form: Reviewer 2

Recommendation

Major revision is needed (please make suggestions in comments)

Do you have any ethical concerns with this paper?

No

Comments to the Author

The manuscript has improved, although it could have been written in a more focused way. There are still unclear formulation in the text, spelling mistakes and misleading citations. Moreover the

quality of most figures needs to be improved.

Here some examples:

“plasma membrane though the translocon” – through

“These 3D imaging techniques allow the visualisation of complexes in their physiological environment associated with native membranes (10, 11) when the structure does not exceed a certain thickness (0.5–1 μm).” – the cited journals do not fit to the message

“The open state can accommodate the unfolded chain and alpha-helix (13).” – please site an original paper, not a review

“TRAP is a ubiquitous protein complex present in all eukaryotes.” – Please cite the original data showing the presence of TRAP in yeast

“TRAP gamma is very close to the ribosomal protein rpl38 and an interaction between is plausible;” ???

Decision letter (RSOB-19-0244.R1)

13-Mar-2020

Dear Dr Russo,

We are writing to inform you that the Editor has reached a decision on your manuscript RSOB-19-0244.R1 entitled "Understanding the mammalian TRAP complex function(s)", submitted to Open Biology.

As you will see from the reviewers' comments below, there are a number of criticisms that prevent us from accepting your manuscript at this stage. The reviewers suggest, however, that a revised version could be acceptable, if you are able to address their concerns. If you think that you can deal satisfactorily with the reviewer's suggestions, we would be pleased to consider a revised manuscript.

The revision will be re-reviewed, where possible, by the original referees. As such, please submit the revised version of your manuscript within four weeks. If you do not think you will be able to meet this date please let us know immediately.

When submitting your revised manuscript, please respond to the comments made by the referee(s) and upload a file "Response to Referees" in "Section 6 - File Upload". You can use this to document any changes you make to the original manuscript. In order to expedite the processing of the revised manuscript, please be as specific as possible in your response to the referee(s).

Please see our detailed instructions for revision requirements
<https://royalsociety.org/journals/authors/author-guidelines/>

Sincerely,
The Open Biology Team
mailto: openbiology@royalsociety.org

Editor's comments to Author:

Please address the comments of the referees and the quality of some of the images used will need improving.

Reviewer(s)' Comments to Author(s):

Referee: 1

Comments to the Author(s)

The author has attended some of the criticisms raised in the previous revision but still some corrections are needed. Following I will describe some specific examples but the manuscript contains more than the mentioned.

Specific:

1- Abstract. The first sentence is incorrect after the revision, since 1/3 of the proteins are not translocated or "reside" into the ER. Actually, this percentage of proteins are translocated or 'inserted' through the ER, but a large population can move to other membranes like Golgi, plasma,..., meaning that not all the membrane proteins inserted into the ER in fact "reside" in the ER.

2- Introduction. The first paragraph of this section is specially confusing. For instance, the first sentence is incorrect, not all prokaryotic membrane proteins are transported across the plasma membrane through Sec A. The second sentence regarding eukaryotic cells must be corrected according to what I mentioned above; and the idea behind the classification in "soluble intracellular, soluble secreted, type I/II..." makes no sense to me if it is referring to one-third of proteins that are targeted to the membrane.

The sentence (line 9) "In mammals,..." it's incorrect in its current form, several truncated forms of strict cotranslationally inserted proteins as short as 86-96 residues have been crosslinked to SRP and/or Sec61alpha. Examples: DOI: 10.1038/nsmb994 doi: 10.1128/JVI.00125-10.

The expression "no-clients of translocons..." in the last sentence of the introduction has no sense to me.

In general, the introduction of this review must be carefully curated, for that the author should read and cite some authorized recent revisions, like doi: 10.1146/annurev-biochem-013118-111717 and doi: 10.1515/hsz-2014-0205, to avoid incorrections or imprecise formulations related to the mechanisms of targeting and cotranslational insertion of membrane proteins.

3- Fig. 1. There is an extra loop between TM2 and TM3 in TRAP gamma that should be removed.
4- Page 14, line 7: replace beta-barrel by beta-sheet.

5- Page 14, the first sentence of the second paragraph is clearly outdated (the reference given is close to 25 years old!). A chaperoning function for TRAM has been suggested and its topology defined, doi: 10.1016/j.jmb.2011.01.009.

6- Reference 61 (page15, line 6) it's not related to what it is mentioned in the text, please remove it.

7- Introduce STRING algorithm on page 15, not every reader should know what this algorithm does.

8- Page 16: add 'expression' between "TRAP induction" in the second and third points suggested for TRAP role in the UPR pathway.

9- Regarding the non-canonical EF domain in *ssr1* on page 16 add '(see below)' at the end of the sentence.

10- Page 17, last sentence, please correct Sec62.

11- Tab. 7: explain panel B, what are the red, gray and green colors referred to? Why there are three sequence lines if only compares TRAP-alpha with Parv?

12- Page 19: in the sentence "The presence of non-canonical EF- hand domain almost confirms that this subunit binds Ca²⁺", 'suggests' is a better choice than "almost confirms".

Referee: 2

Comments to the Author(s)

The manuscript has improved, although it could have been written in a more focused way. There are still unclear formulation in the text, spelling mistakes and misleading citations. Moreover the quality of most figures needs to be improved.

Here some examples:

"plasma membrane though the translocon" - through

"These 3D imaging techniques allow the visualisation of complexes in their physiological environment associated with native membranes (10, 11) when the structure does not exceed a certain thickness (0.5-1 µm)." - the cited journals do not fit to the message

"The open state can accommodate the unfolded chain and alpha-helix (13)." - please site an original paper, not a review

"TRAP is a ubiquitous protein complex present in all eukaryotes." - Please cite the original data showing the presence of TRAP in yeast

"TRAP gamma is very close to the ribosomal protein rpl38 and an interaction between is plausible;" ???

Author's Response to Decision Letter for (RSOB-19-0244.R1)

See Appendix B.

Decision letter (RSOB-19-0244.R2)

23-Apr-2020

Dear Dr Russo

We are pleased to inform you that your manuscript entitled "Understanding the mammalian TRAP complex function(s)" has been accepted by the Editor for publication in Open Biology.

Sincerely,
The Open Biology Team
[mailto: openbiology@royalsociety.org](mailto:openbiology@royalsociety.org)

Appendix A

Referee 1

- 1, 2 – Sentences have been rephrased and language improved.
- 3 – The reference has been added (originally was UniProt but I found a publication).
- 4 – TRAM 1/ 2 has been removed, only TRAM is mentioned in all paper.
- 5 – This sentence is changed with the following: However, the mature protein plays also a role (29, 30); indeed, no-clients of translocons even by adding a SP are rejected (31-33). (last two lines of the Introduction).
- 6 - The yellow, green, and black (figure) is explained.
- 7 – Fig. 3: no data are present about TRAP subunits PBD structure; I used RaptorX for prediction and also I included the previous fig.3 which is important to understand how the complex is localized in the ER membrane.
- 8 – The Tab. Screenshots have been considerably improved and the program Geneious that I used is mentioned. The transmembrane domains are shown in all protein alignments and in the legend the different domains are specified. Moreover, there is the figure 1 that shows these domains.
- 9 – In the paragraph is mentioned type I membrane protein (first sentence).
- 10 – Fig. 4, I added to the cartoon the rpl38 protein.
- 11 – I have written more general sentences because I cannot explain precisely what is the difference between TRAP and TRAM.
- 12 - The right reference has been written.
- 13 – The literature has been revisited.
- 14 - The figures have been removed. The STRING consortium is mentioned at the end of the paper (there is written about STRING prediction in the text), (also RaptorX).
- 15 – Tab. Corrected.
- 16 – This was a mistake I copied the text from another my document.
- 17 - Sec62 as well Sec63 are described in the introduction.
- 18 – Tab. 8 refers to last paragraph and it is mentioned.

Referee 2

Shortly I introduced the translocon of prokaryotes, the first sentences (reference 1).

All references have been revisited.

Sentences have been rephrased and language improved.

Some figures have been removed: 1, 6, 7. The tables considerable improved.

The molecular weight of TRAP has been corrected (about 90 Kda).

I added the alignment of RNA of the different isoforms. However, I did not find data about mRNA, what I show is retrotranslation from protein sequences. I do not know if it is worth to show these RNA alignments. (?)

Many thanks

Appendix B

Referee 1

1. Abstract: first sentence changed.
2. Introduction: revised all introduction, removed superfluous sentences and corrected some sentences. The sentence “no-clients of the translocon...” has been changed to explain that the sequence of the mature protein is also relevant for translocation.
3. Figure 1: extra loop removed.
4. Replaced beta -barrels with beta-sheets, actual page 15.
5. Reference removed and added the new reference.
6. Reference removed.
7. String introduced.
8. “Expression” is added.
9. “see below” is added.
10. I did not understand this point...and did not make chances.
11. Table (B, non-canonical EF-hand domain alignment) is cut, no extra lines.
12. Changed the verb with suggest.

Referee 2

Many superfluous sentences have been removed, probably it is a more focused review paper. Spelling corrected (I hope completely).

Citations corrected; figures improved.

References 10-11 removed and added the proper reference.

Reference 13 removed and added another reference.

TRAP reference that contains also data about yeast is specified.

TRAP gamma and rpl38 are very closed by cryo-EM data, but the sentence “an interaction between them is plausible” has been removed.

Thank you very much for your very constructive reviews.